# Study of the Macro and Micro Characteristics of and Their Relationships in Cemented Backfill Based on SEM

**DOI:** 10.3390/ma16134772

**Published:** 2023-07-01

**Authors:** Fengwen Zhao, Jianhua Hu, Taoying Liu, Tan Zhou, Qifan Ren

**Affiliations:** 1School of Resources and Safety Engineering, Central South University, Changsha 410083, China; dr_zfw94@163.com (F.Z.); taoying@csu.edu.cn (T.L.); zt3153@outlook.com (T.Z.); 2Zijin School of Geology and Mining, Fuzhou University, Fuzhou 350108, China; 3CERIS, Instituto Superior Técnico, University of Lisbon, Av. Rovisco Pais 1, 1049-001 Lisboa, Portugal; qifanren@tecnico.ulisboa.pt

**Keywords:** backfill material, slit island method, scanning electron microscopy image, macro and micro characteristics, functional relationship

## Abstract

Nuclear magnetic resonance can only quantitatively obtain porosity and pore size distribution, but as a conventional microstructure observation technology, scanning electron microscope (SEM) can select different magnifications to observe the microstructure of backfill materials. However, the processing of SEM images is not deep enough. In this paper, Image-Pro Plus 6.0 software was used to extract the data from SEM images, and the parameters of the area, and the perimeter, aspect ratio and roundness of the pores in the SEM images were obtained. The fractal characteristics of the pores in the SEM image were obtained by using the slit island method fractal theory. The concretization and quantification analysis of the pores’ complexity were achieved. A functional relationship model for the strength and pore parameters was constructed; thus, the influence law of pore characteristics on strength was quantitatively analyzed. The conclusions included: (1) Pore parameters indicate that most pores in backfill are irregular, and only a few pores are regular—however, the whole structure has good fractal characteristics (R^2^ > 0.96). (2) The fractal dimension of pores is directly proportional to the roundness, the aspect ratio, and the pore content of pores—which indicates that the complexity of pores is related to both pore shape and pore content. (3) The strength had a linear inverse relationship with the roundness, aspect ratio, pore content, and fractal dimension—which indicates that all characteristics of pores have a certain influence on the strength.

## 1. Introduction

Uniaxial compressive strength (UCS) is an important basis for the availability of backfill materials in mines [1,2,3]. However, there are many factors affecting UCS—among which, pore characteristics are the main factors [3,4,5]. Therefore, it is very critical to study pore characteristics. In previous studies, the methods for studying pores have mainly included the mercury intrusion method, nuclear magnetic resonance (NMR) technology, and scanning electron microscope (SEM) technology, etc.—in which SEM can select different magnifications for observation as needed [6,7,8]. Huafu Qiu et al. used SEM to magnify the hydration products in backfill by 5000 times for observation, so as to qualitatively analyze the types of hydration products in the backfill with different waste rock contents [6]. Jianhua Hu et al. used SEM to magnify the pore structures in backfill by 500 times for observation, and used binaryzation processing so as to qualitatively analyze the pore distribution in the backfill [7]. In order to further study SEM images, some scholars have proposed fractal theory—as such, box dimensions can be used for the quantitative analysis of SEM images [7,9,10]. Fengwen Zhao et al. used box dimensions to process SEM images of backfill materials to analyze their pore characteristics [9].

Fully obtaining SEM image information is helpful for comprehensively understanding pore characteristics. In order to further fully analyze SEM images, some scholars have used Image-Pro Plus 6.0(IPP) software for analysis [11,12,13,14]. IPP software can obtain many parameters of graphics in an image, such as the perimeter, area, diameter, and other parameters. Xiaojie Yang et al. used IPP to obtain the average pore size, area, and other parameters of rock pores [14]. Chunling Zhong et al. used IPP to obtain pore parameters such as pore size percentage, pore area ratio, average pore size, and so on, of concrete [15]. Through research, it has been found that there is a certain functional relationship between the perimeter and area of the graphics, which can be studied using the fractal theory of the slit island method, so as to obtain the fractal characteristics of the graphics [16,17,18,19,20]. Apostolos Papadopoulos et al. used the slit island method to analyze the pore roughness of soil, so as to obtain the pore fractal characteristics of soil [16]. Zhibao Zhu used the slit island method to analyze a rock’s SEM image and obtained the pore fractal characteristics of rock [20]. However, the use of IPP 6.0 software and slit island method analysis are less used in the backfill materials field. Therefore, applying them to research on backfill materials could provide greater assistance in research on backfill materials.

Based on the basis and shortcomings of the above research, this study used tailings and tailing sludge as aggregates, Portland cement and fly ash as cementing materials, and lime as an alkaline activator to make backfill materials. The UCS and pore structure of the backfill material were obtained by testing. IPP 6.0 software was applied to SEM images of the backfill material to obtain the pore parameters, and the slit island method was used to further analyze the pore parameters, so as to obtain the fractal characteristics of the pores—thus achieving a quantitative analysis of the SEM images. Therefore, this method was used to study the influence law of pore characteristics on UCS.

## 2. Materials and Methods

### 2.1. Materials

Phosphogypsum comes from the phosphorus chemical plant of Sanning Mining Company, Hubei, China; fly ash from the coal chemical plant of Sanning Mining Company, Hubei, China; and tailings and tailing sludge came from the concentration plant of Sanning Mining Company, Hubei, China. Ordinary Portland cement was purchased in Changsha cement market, Changsha, China. Lime was commercially available and contained 85% CaO. The experimental water was tap water (daily drinking water for residents in Changsha, China). The above experimental materials were analyzed by the sieving method, a laser particle size analyzer, X-ray fluorescence spectroscopy (XRF), and X-ray diffraction (XRD), and the particle size distribution (Figure 1 and Figure 2), physical properties, and mineral composition (Table 1 and Table 2) were determined. The difference in packing density and apparent density in Table 1 is the calculation method for the material volume.

### 2.2. Experimental Methods

#### 2.2.1. Scheme Design

Among the raw materials, the tailings and tailing sludge without cementitious properties were used as aggregates. Ordinary Portland cement and fly ash with cementitious properties were used as cementing materials. Lime was used as an alkaline activator. The manufacturing and curing process for the backfill sample were as follows: standard samples were made from slurries of different proportions using cylindrical molds (50 mm × 100 mm); these were labeled as B and A0–A6 in sequence according to the proportions. Three specimens in each group were prepared; thus, there were a total of 24 specimens. The specific proportions are shown in Table 3. In this table, tailing sludge: tailings = 1:3, fly ash: tailing sludge = 1:2. The samples were put into the curing box for curing. All tests were carried out on the 28th day. The experimental process is shown in Figure 3.

#### 2.2.2. Sample Testing

UCS test: The No. 01000405 pressure testing machine made by Changlu, Ji’nan Zhongluchang Testing Machine Co., Ltd., Jinan, China, was used for the UCS test. The loading method was constant force (0.2 kN/s), and the placement position of the sample was at the center of the pressure plate.SEM test: A JSM-IT500LV SEM manufactured by JEOL Ltd. was used. Samples were taken at the center of the damaged backfill sample after UCS. Samples, after being sprayed with gold, were observed by SEM.NMR test: A MesoMR23-060H NMR instrument made by the Suzhou Niumai Company, Suzhou, China, was used. The test samples were saturated samples. After setting the parameters of the instrument, the samples were then placed in the coil for testing.

## 3. Result

### 3.1. UCS Characteristics

The strength of the backfill material was tested according to the above steps, and the results are shown in Table 4. The change rate refers to the percentage of difference value between each group of data and the control group; the calculation formula is shown in Equation (1). In Table 4, the change rate of Group A0 is relative to Group B and the change rate of A1 to A6 is relative to Group A0. As can be seen from Figure 4, the UCS of Group A1 is the largest, and the change rate of this group is also the largest among the positive values. The change rate of Group A0 was positive and the UCS increased slightly; this shows that adding a certain amount of phosphogypsum and lime could supplement its strength. The change rates of the latter five groups were negative, and their strength decreased—indicating that adding too much lime is harmful to strength. There are two reasons for this phenomenon: On the one hand, as an alkaline activator, lime can activate the pozzolanic effect of phosphogypsum and fly ash, and participate in the hydration reaction, which is beneficial to strength [21,22]. On the other hand, the volume of lime expands after absorbing water, resulting in micro-cracks—thus the strength is reduced [23].
(1)$$∆\sigma =\frac{\sigma_{i}-\sigma_{c}}{\sigma_{c}}\times 100\%$$
where ∆σ represents the change rate, $\sigma_{i}$ represents each group of UCS, and $\sigma_{c}$ represents the UCS of the control group.

### 3.2. NMR Characteristics

NMR was mainly used to test the pore size distribution and content of the samples [9]. Figure 4 shows the NMR characteristics of the pores of the backfill material sample (taking Group B as an example), and Figure 5 shows the contents of the various pores in all groups of samples. According to existing research [9], T_2_ is used to characterize the pore size (the larger the T_2_, the larger the pore size). From Figure 4, it can be seen that pores can be divided into two types: namely, small pores and big pores. The sum of the small pore content (P_S_) and the big pore content (P_B_) is the porosity (P_A_). According to the NMR data, the porosity ranged from 4.38 to 5.37%, with the content of the small pores accounting for over 89%—which were the main pores. As can be seen from Figure 5, the overall porosity showed a trend of first decreasing and then increasing. This trend indicates that the amount of lime added needs to be appropriate to produce the best results.

### 3.3. SEM Characteristics

SEM can achieve the purpose of observation through different magnifications, and a low magnification can be used to observe the pore distribution characteristics in a material. The distribution and morphology of pores can be clearly observed by processing SEM images. Figure 6 shows the SEM image of Group B magnified 500 times and its processed image. Figure 6a is the original SEM image, Figure 6b is Threshold image, and Figure 6c is the binaryzation image of the SEM image. The binarization principle follows the following principles; the binarization formula is [24,25]:(2)$$f=\left\{\begin{array}{cc} 0 & f<T\\ 1 & f\geq T \end{array}\right.$$
where *T* is the gray threshold, 0 represents the solid phase—i.e., black—and 1 represents the pore, i.e., white. It can be seen from Figure 6 that the pore shapes were all different, with a small number of regular shapes and mostly irregular shapes. From the perspective of pore distribution, big pores were relatively concentrated and small pores were relatively dispersed.

## 4. Analysis and Discussion

### 4.1. Extraction of Pore Parameters

Image-Pro Plus is an image processing software that can extract various parameters of pores in an image, and can also limit the parameters according to specific requirements. According to relevant research [26,27], when the pore size is greater than 0.1 μm, the pore results in great damage to strength. Therefore, this study only focuses on pore sizes greater than 0.1 μm for parameter extraction. In this study, the parameter extraction of the pores included their perimeter, area, roundness, and aspect ratio. The parameter extraction process is shown in Figure 7. The aspect ratio, also known as the length–width ratio (for pores), refers to the ratio of the longest diameter of the pore to the longest diameter perpendicular to it. Roundness, also known as the complexity, refers to the similarity between the pore shape and circle. Their expressions are as follows [28]:(3)$$d=\frac{L}{B}$$
(4)$$r=\frac{P^{2}}{4\times \pi \times A}$$
where *d* represents the aspect ratio, *L* represents the longest diameter, *B* represents the longest diameter perpendicular to *L*, *r* represents the roundness, *P* represents the perimeter, and *A* represents the area. According to Equations (3) and (4), the critical values of d and r for conventional pore shapes (circle, square, regular triangle) can be calculated. The smaller the d is (the minimum is 1), the closer to square the pore is, and the larger the d is, the slenderer the pore is. The smaller the r (the minimum is 1), the more regular the pore is, and the larger r, the more irregular the pore is. When *r* = 1, the pore shape is equivalent to a circle; when *r* = 1.27, the pore shape is equivalent to a square; when *r* = 1.65, the pore shape is equivalent to a regular triangle; when *r* > 1.65, the pore shape is equivalent to a polygon.

Figure 8 shows the distribution of the pore roundness and aspect ratio. It can be seen from Figure 8 that most of the pore shapes are irregular and that the pores are complex. The average roundness was 2.41 and the average aspect ratio was 1.92. Figure 9 shows the pore shape values of each group of samples. It can be seen from Figure 9 that the range and the dispersion of the roundness were all large, while the range and the dispersion of the aspect ratio were all small. As the groups changed, the values of the roundness and aspect ratio showed a trend of first decreasing and then increasing.

### 4.2. Slit Island Method Fractal Analysis

#### 4.2.1. Slit Island Method

The slit island method, also known as the Area–Perimeter method, is mainly used to analyze the fractal characteristics of particles and pores. According to relevant research [28,29,30], there is a certain functional relationship between the perimeter and area of a graph. For plane pores, their perimeter and area also conform to this law. That is, the relationship between the perimeter and area conforms to the following formula:(5)$$P^{1/D}\propto A^{1/2}$$

Equation (5) can be converted into:(6)$$P^{1/D}=C\times A^{1/2}$$

Taking logarithms on both sides of Equation (6), we get:(7)$$\ln\left(P\right)=\frac{D}{2}\ln\left(A\right)+\mathrm{DlnC}$$

According to Formula (7), when ln (*P*) and ln (*A*) conform to a linear relationship, the two-times of the slope is the fractal dimension of the slit island method. For SEM images, the process of solving the fractal dimension of the slit island method is shown in Figure 10. Firstly, the SEM image is converted into a binaryzation image, and then, the perimeter and area of the pore are obtained by Image-Pro Plus. Finally, ln (*P*) and ln (*A*) are fitted according to Formula (7) to solve their slope. The two-times of the slope is the fractal dimension of the slit island method.

#### 4.2.2. Fractal Features

According to the above fractal principle, each group of samples was analyzed, and the results are shown in Figure 11. It can be seen from Figure 11 that the samples in each group had good fractal characteristics (R^2^ > 0.96), indicating that the pores inside the samples showed good self-similarity. The fractal dimension of pores reflects the complexity of the pores—that is, the larger the fractal dimension, the more irregular and complex the pores are [31,32]. Through a comparative analysis of Group B and Group A0, it can be seen that the pores in Group B are more complex than those in Group A0. This is because phosphogypsum was added in Group A0, which can effectively fill the pores inside the sample, resulting in a reduction in pore volume and pore size—thus reducing the complexity level. Through a comparative analysis of Group A0 and Group A1, it can be seen that the pores in Group A0 were more complex than those in Group A1; this is because lime was added in group A1, which can promote the chemical reactions of phosphogypsum and fly ash, so as to increase the amount of hydration products. These hydration products can fill the pores, resulting in further reductions in pore volume and smaller pore sizes, and so the complexity is lower. By comparing and analyzing Groups A2 to A6, it can be seen that the pore complexity had an increasing trend; its volume expanded after the excess lime absorbed the water, increasing the internal microcracks in the sample [23] and resulting in increases in pore volume—thus, the complexity level increased.

### 4.3. Construction of a Functional Relationship Model

#### 4.3.1. Relationship between the Dimension and Pore Characteristics

The fractal dimension reflects the complexity of pores. The porosity reflects the size of pores and the roundness and aspect ratio reflect the shape characteristics of pores. Therefore, there is a certain functional relationship between the fractal dimension and pore characteristics. The functional relationships between the fractal dimension and roundness, aspect ratio, and pore content were established, and the results are shown in Figure 12. It can be seen from Figure 12 that the functional relationships between them were good (R^2^ > 0.79) and conformed to the following functional relationship:(8)$$D_{I}=1.14\times 10^{-5}\times e^{3.41r}+1.45$$
(9)$$D_{I}=0.20d+1.10$$
(10)$$\begin{array}{c} P_{A}=13.15D_{I}-14.76\\ P_{S}=10.89D_{I}-11.84 \end{array}$$
where *D_I_* represents the fractal dimension, *r* represents the roundness, and *d* represents the aspect ratio. From the above formulas, it can be found that the more irregular the pore shape is and the larger the pore is, the more complex the pore is. When the roundness reached a certain value, it had a great influence on the fractal dimension—showing an exponential increasing relationship. The aspect ratio and pore content were linearly, directly proportional in relation to the fractal dimension—indicating that the more slender the pore is and the larger the pore is, the more complex it is.

#### 4.3.2. Relationship between UCS and Pore Characteristics

As important parameters of a pore, the pore shape and porosity have a significant impact on strength. Therefore, it is of great significance to analyze the influence of pore characteristics on strength. The functional relationships between strength and pore characteristics were established, and the results are shown in Figure 13. It can be seen from Figure 13 that the functional relationships between them were good (R^2^ > 0.8) and conformed to the following functional relationship:(11)σ=−1.43r+5.52
(12)σ=−2.91d+8.08
(13)$$\begin{array}{c} P_{A}=-0.89\sigma +6.87\\ P_{S}=-0.74\sigma +6.09 \end{array}$$
where *σ* represents the UCS (MPa). From the above formulas, it can be found that the more irregular the pore is and the larger the pore is, the lower the strength is. This is because the more irregular the pores, the more irregular the solid phase and the weaker its supporting effect—thus, the strength is reduced. From the relationship between the aspect ratio and strength, it can be seen that the slenderer the pore is, the lower the strength is. This illustrates that slender and long pores result in greater damage to strength than short and coarse pores; this is because the more slender and the longer the pores are, the easier it is to form connected pores, and the more easily the sample can be damaged—thus, the strength is reduced.

#### 4.3.3. Relationship between UCS and Dimension

Pore characteristics are important factors affecting strength, and the fractal dimension reflects the overall characteristics of pores. Therefore, it is necessary to use the fractal dimension to quantitatively analyze strength. The functional relationship between strength and the fractal dimension was established, and the result is shown in Figure 14. It can be seen from Figure 14 that the functional relationship between them was good (R^2^ = 0.89) and conformed to the following functional relationship:(14)$$\sigma =-13.07D_{I}+21.78$$

It can be seen from the above formula that the larger the fractal dimension, the lower the strength. This illustrates that the more complex the pores are, the lower the strength is. This conclusion is consistent with the results of Lang Liu et al.’s study on the influence law of pores on strength [27]. This is because the more complex the pores are, the more irregular and slender the pores are—resulting in the weakening of the supporting effect of the solid phase and the enhancement of the pores connectivity, and thus reducing the strength.

## 5. Conclusions

Pore characteristics are important factors affecting UCS. It is very important to comprehensively analyze the influence law of pore characteristics on UCS. In this study, the UCS and pore structure of backfill material were obtained by a pressure-testing machine, NMR, and SEM. Additionally, IPP 6.0 software was used to obtain the pore parameters of the SEM images, and the slit island method was used to further analyze the fractal characteristics of pores so as to achieve the concretization and quantification analysis of pore complexity, and then to construct a functional relationship model between strength and pore characteristics. The results were as follows:
The pores in the SEM image were complex; this could be seen from the roundness of most of the pore shapes (>80%) being irregular (*r* > 1.65). However, the whole material showed good fractal characteristics (R^2^ > 0.96); the fractal dimension ranged from 1.450 to 1.558.The complexity of pores is related to both pore shape and pore content. The more irregular the pores are and the larger the pores are, the larger the fractal dimensions of the pores are. That is, the fractal dimension of pores is exponentially, directly proportional to the roundness of the pores (i.e., *D_I_ = a*e^br^ + c*), and linearly, directly proportional to the aspect ratio of the pore and pore content (i.e., *D_I_ = kx + b*, *k* > 0).There are many factors that affect UCS; these include not only the pore content but also the pore shape characteristics. It can be seen from UCS functional relationship model that the UCS had a linear inverse relationship with the roundness, aspect ratio, pore content, and fractal dimension (i.e., *σ = kx + b*, *k* < 0).The addition of lime should be in moderation. An appropriate amount of lime can increase hydration products, which is beneficial to backfill, but excessive lime is harmful to backfill.

## Figures and Tables

**Figure 1 materials-16-04772-f001:**
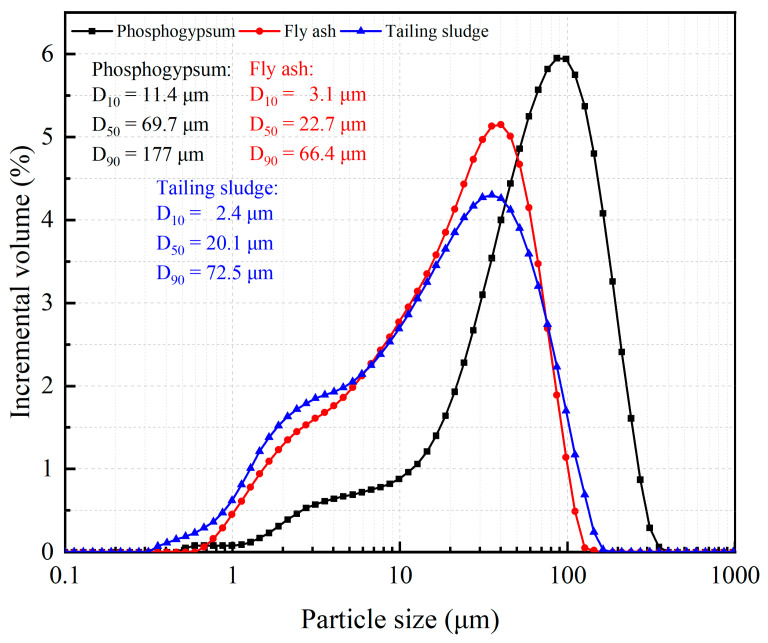
Particle size distribution of phosphogypsum, fly ash, and tailing sludge.

**Figure 2 materials-16-04772-f002:**
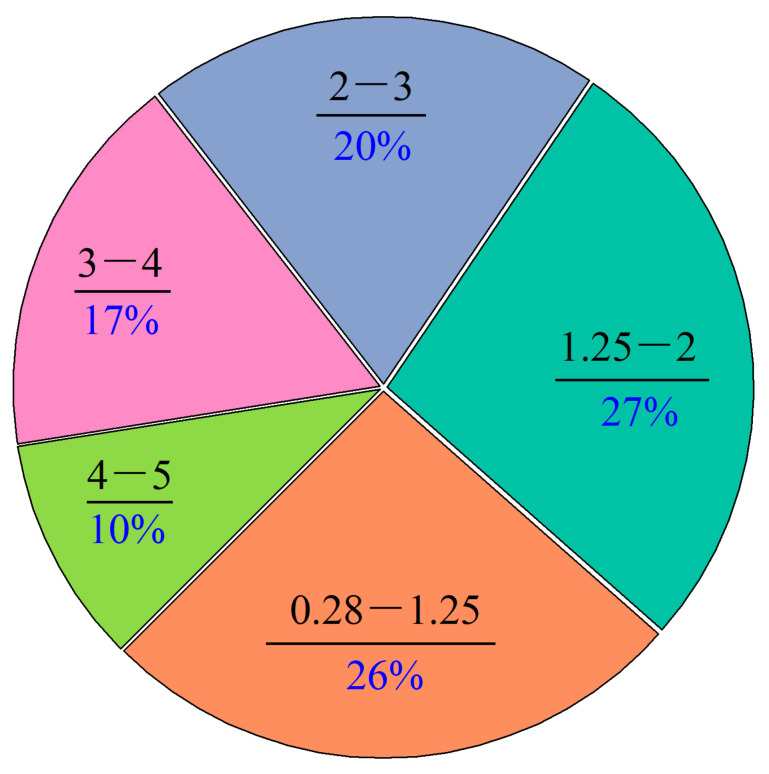
Particle size distribution of tailings (mm).

**Figure 3 materials-16-04772-f003:**
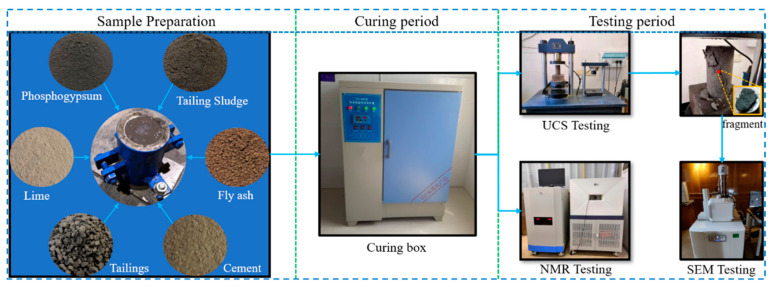
Experimental flowchart.

**Figure 4 materials-16-04772-f004:**
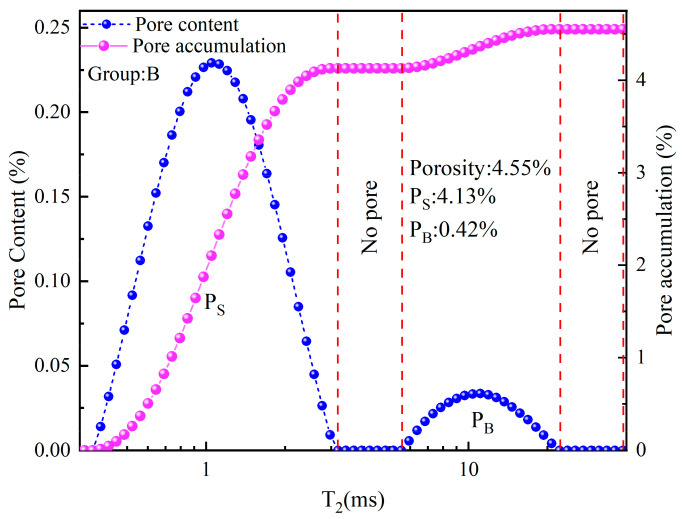
NMR characteristic of the pores of the backfill material sample.

**Figure 5 materials-16-04772-f005:**
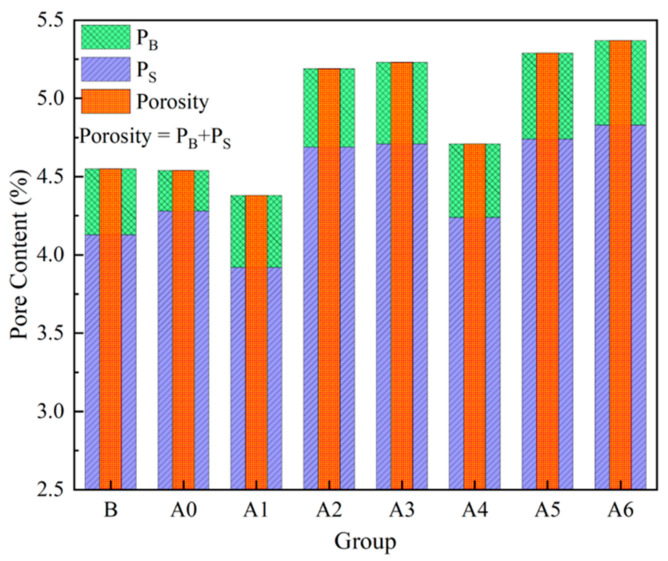
The contents of various pores of all groups of samples.

**Figure 6 materials-16-04772-f006:**
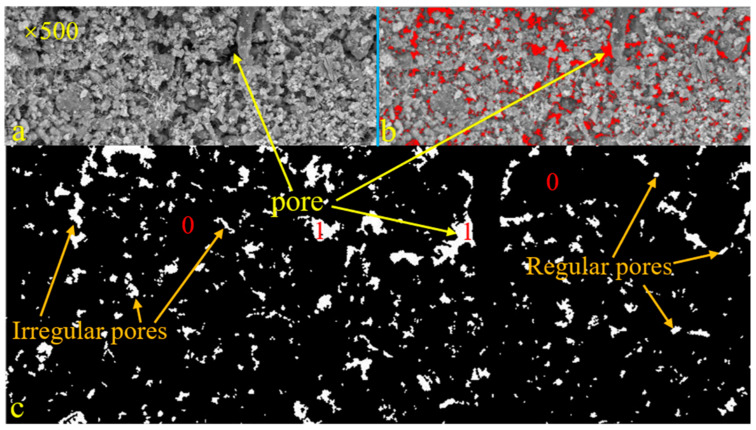
SEM image of Group B magnified 500 times and its processed image (**a**. the original SEM image, **b**. Threshold image, **c**. the binaryzation image of SEM image).

**Figure 7 materials-16-04772-f007:**
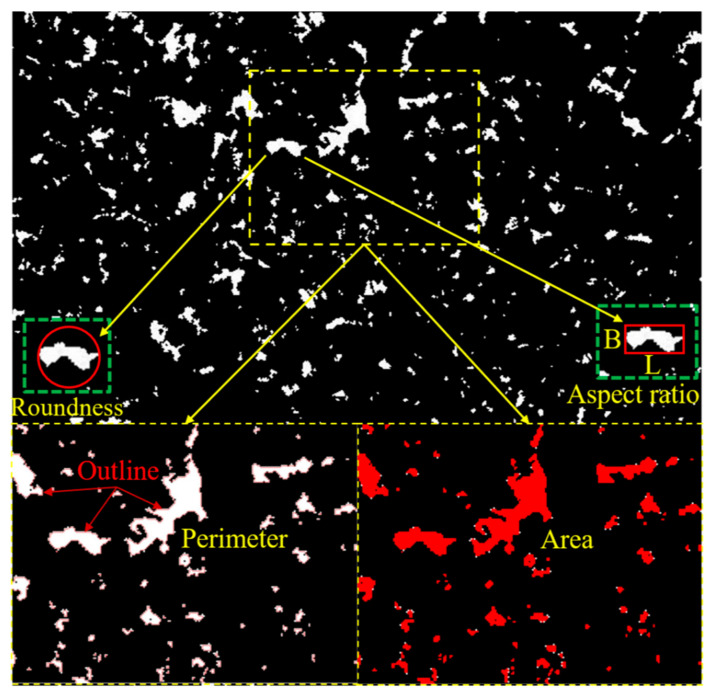
Acquisition of pore perimeter and area (*L*—the longest diameter, *B*—the longest diameter perpendicular to *L*).

**Figure 8 materials-16-04772-f008:**
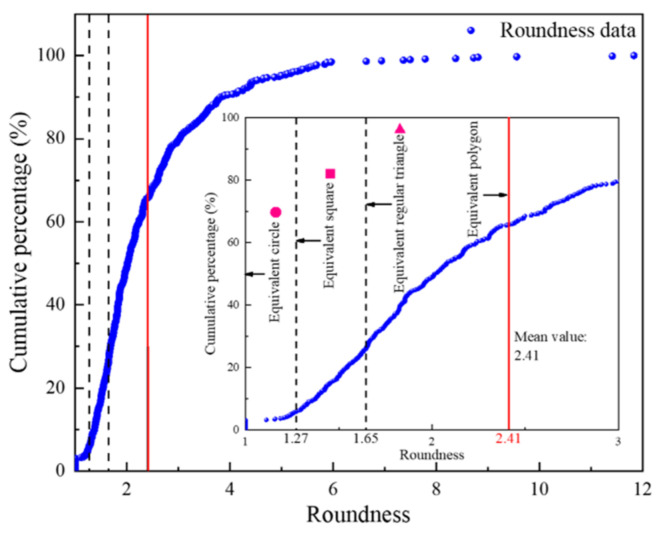
Distribution characteristics of pore roundness and aspect ratio.

**Figure 9 materials-16-04772-f009:**
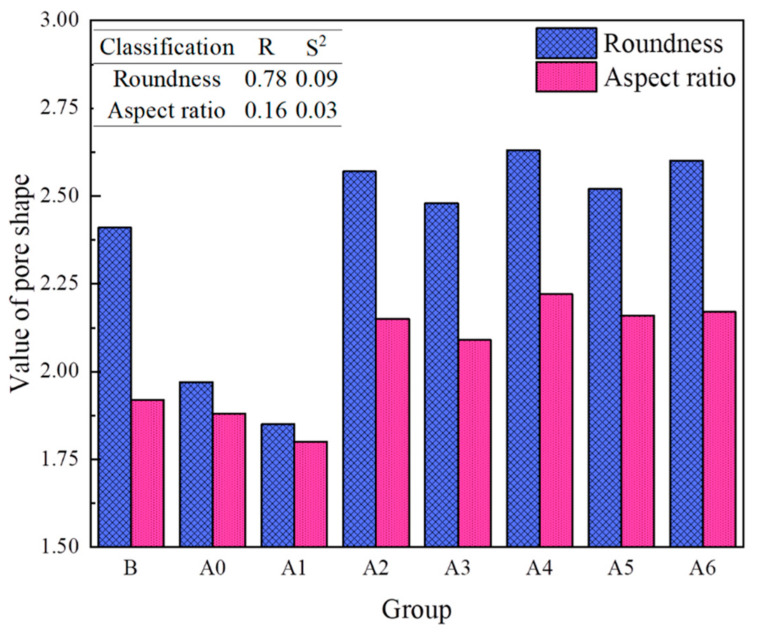
Pore shape values of each group of samples.

**Figure 10 materials-16-04772-f010:**
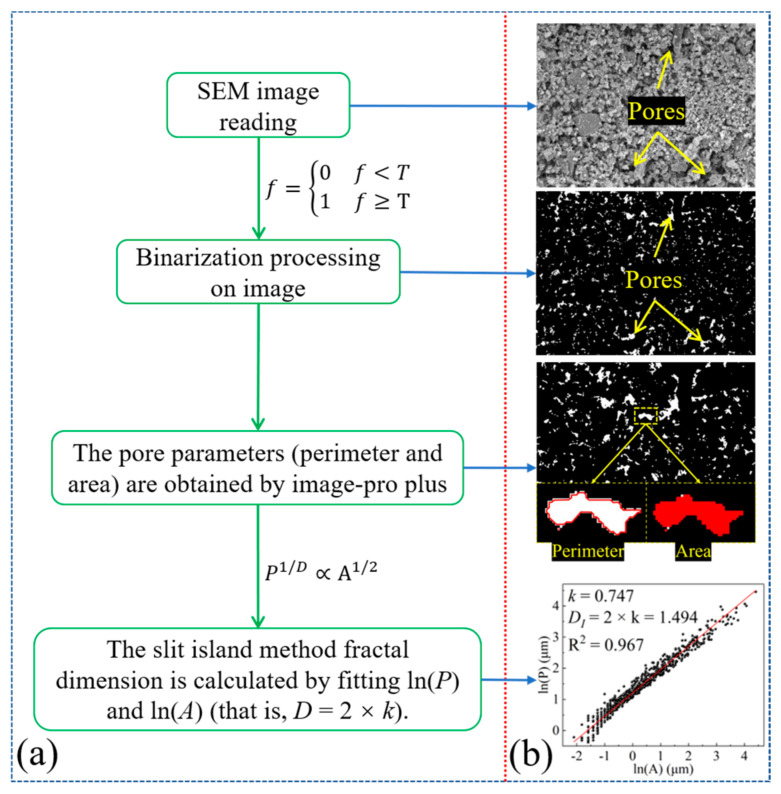
The solution process of the slit island method fractal dimension for SEM images (**a**. The solution process of fractal dimension, **b**. Solution schematic image of fractal dimension).

**Figure 11 materials-16-04772-f011:**
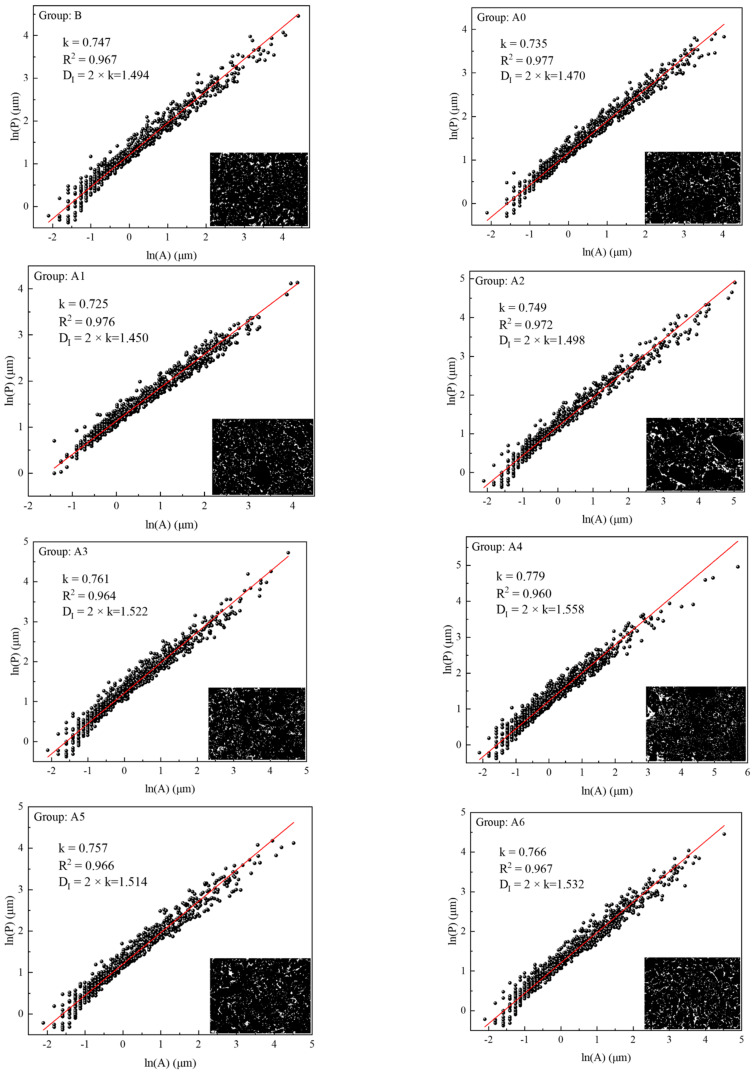
Fractal characteristics of each group’s sample. The magnification: ×500.

**Figure 12 materials-16-04772-f012:**
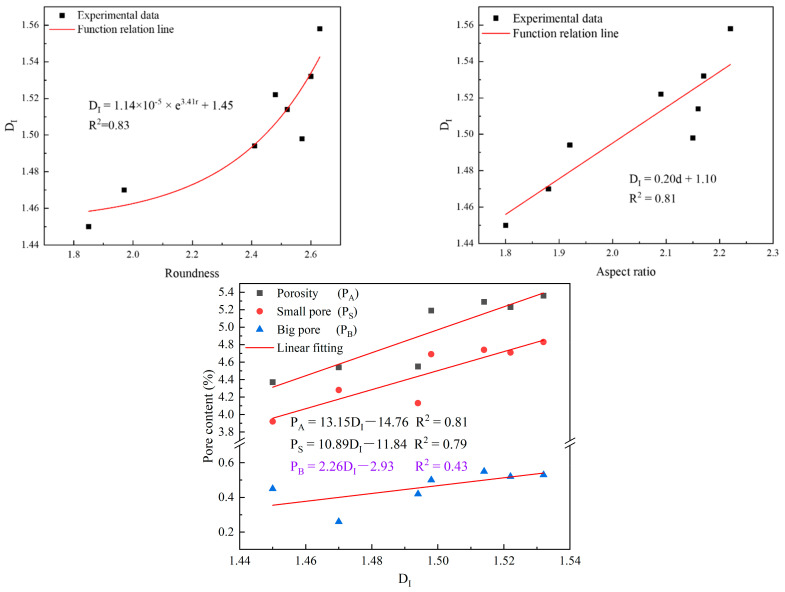
Functional relationship between the fractal dimension and pore characteristics.

**Figure 13 materials-16-04772-f013:**
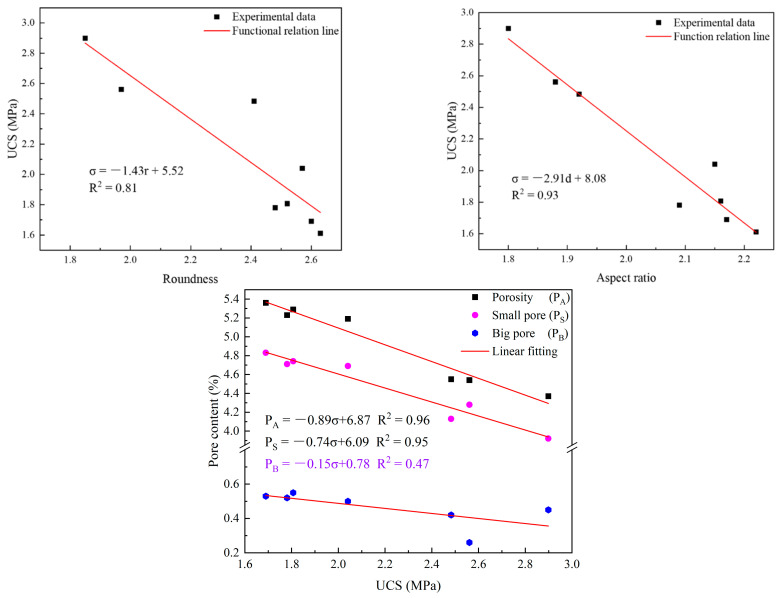
Functional relationship between UCS and pore characteristics.

**Figure 14 materials-16-04772-f014:**
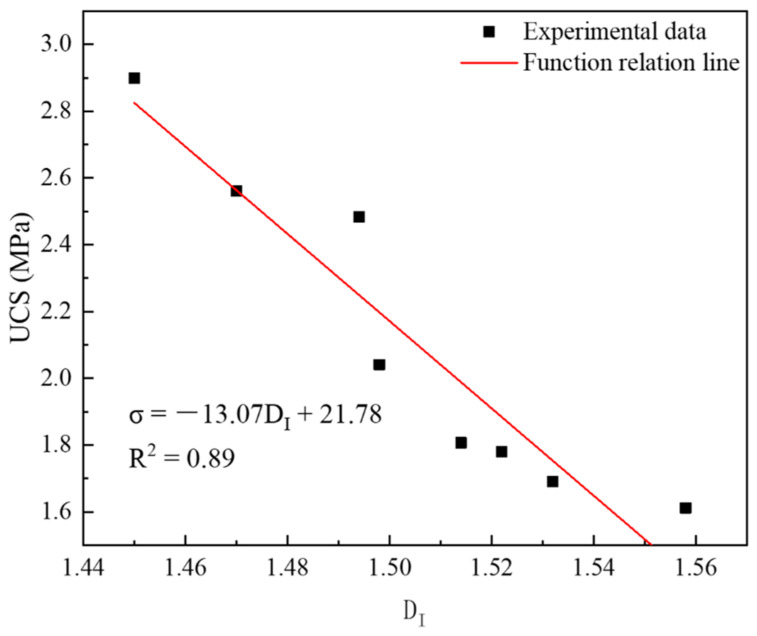
Functional relationship between UCS and fractal dimension.

**Table 1 materials-16-04772-t001:** Physical properties of test materials.

Sample	Apparent Density/(kg∙m^−3^)	Packing Density/(kg∙m^−3^)	Surface Moisture Content/%
Tailings	2626	1464	0.120
Tailing sludge	2653	923	0.974
Fly ash	1990	650	0.049
Phosphogypsum	1992	850	8.11

**Table 2 materials-16-04772-t002:** Mineral composition of the test materials.

Sample	Mass Fraction/%										
	Hydroxyllapatite	Quartz	Hematite	Albite	Plagioclase	Muscovite	Illite	Dolomite	Plaster	Amphibole	Calcite
Tailings	10.15	6.91	12.75	−	-	-	-	69.65	0.2	-	0.34
Fly ash	-	61.55	1.46	15.99	-	20.99	-	-	-	-	-
Phosphogypsum	-	1.35	-	-	-	-	4.05	-	94.02	0.59	-
Tailing sludge	60.94	2.24	-	8.38	11.42	-	9.76	6.30	0.34	-	0.61

**Table 3 materials-16-04772-t003:** Experimental proportion.

Group	Phosphogypsum Content/%	Lime Content/%	Mass Percentage	Cement Tailings Ratio
B	0	0	80%	1:6
A0	20	0		
A1	20	0.2		
A2	20	1		
A3	20	1.8		
A4	20	2.6		
A5	20	3.4		
A6	20	4.2		

**Table 4 materials-16-04772-t004:** Strength characteristics of different groups.

Group	B	A0	A1	A2	A3	A4	A5	A6
UCS/MPa	2.48	2.56	2.90	2.04	1.78	1.61	1.81	1.69
Change rate/%	0	3.14	13.20	−20.30	−30.46	−37.06	−29.44	−34.01

## Data Availability

The data presented in this study are available on request from the corresponding author.

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
