# Peer review of "Study of the Macro and Micro Characteristics of and Their Relationships in Cemented Backfill Based on SEM"

_materials, 2023, doi:10.3390/ma16134772_

Round 1

Reviewer 1 Report

This study is investigating the macro and micro characteristics and their relationships of cemented backfill based on a scanning electron microscope (SEM).  The following notes are to be considered:

1- The abstract is very poorly written and doe not explain the content of the paper.  Besides the first paragraph in the abstract is not needed.  The author should directly start with his work without an introduction.

2- The introduction needs further work and more references are required.

3- In Table 1 discuss the difference in packing density and apparent density.

4- Figure 3 is not clear to the reader

5- Figure 11 is not clear to the reader

The English language needs little improvement

Reviewer 2 Report

My comments are marked on the main text. My main concern is related to discussion. There is no a real discussion of the results. 

English must be checked by an English-speaker

Reviewer 3 Report

The discussion in this work is appropriate and the scientific contribution clear. There are only small changes that could improve the manuscript.

1. In Fig. 3, page 4, levels in red are difficult to read. Please, change the color to white or other.

2. In Figure 4, there are two zones with No big pores. One is already delimited with dashed red lines. Please, indicate the other as well.

1. Consider reviewing the writing. For example, Page 4: “…group, The…”; “…the figure…”; “…B, and…”. Page 5: “…sample[3,5].Fig-, porosity(PA), and so forth.

Page 10: “increasing the micro cracks internal the sample” could be written as “increasing the internal microcracks of the sample

Reviewer 4 Report

Dear Aurors

In my opinion, a correction should be made, page 7 of the publication, a space should be introduced between the numerical value and the unit,

This should also apply to other entries of this type,

The effect of lime content (subsection 4.2.2) and the amount of hydration products seems to be extremely interesting, no reference in the conclusions

Reviewer 5 Report

The study is interesting. Just in case it would be convenient for the authors to better explain the novelty of the study.

I have not got significant modifications for this manuscript.

Round 2

Reviewer 2 Report

All my comments have been addressed. The manuscript is suitable for its publication. Congrats to the authors.